# Prevalence of Bacterial and Protozoan Pathogens in Ticks Collected from Birds in the Republic of Moldova

**DOI:** 10.3390/microorganisms10061111

**Published:** 2022-05-27

**Authors:** Alexandr Morozov, Alexei Tischenkov, Cornelia Silaghi, Andrei Proka, Ion Toderas, Alexandru Movila, Hagen Frickmann, Sven Poppert

**Affiliations:** 1Center of Research of Biological Invasions, Institute of Zoology, MD-2012 Chisinau, Moldova; proka1994@gmail.com (A.P.); iontoderas@yahoo.com (I.T.); amovila@iu.edu (A.M.); 2Natural Geography Department, Shevchenko Transnistria State University, MD-3300 Tiraspol, Moldova; entsdmr@mail.ru; 3Comparative Tropical Medicine and Parasitology, Ludwig-Maximilians-Universität München, 80802 Munich, Germany; cornelia.silaghi@fli.de; 4Institute of Infectology, Friedrich-Loeffler-Institute, 17493 Greifswald, Germany; 5Department of Biomedical Sciences and Comprehensive Care, Indiana University School of Dentistry, Indianapolis, IN 46202, USA; 6Department of Microbiology and Hospital Hygiene, Bundeswehr Hospital Hamburg, 20359 Hamburg, Germany; frickmann@bnitm.de; 7Institute for Medical Microbiology, Virology and Hygiene, University Medicine Rostock, 18057 Rostock, Germany; 8Bernhard Nocht Institute for Tropical Medicine Hamburg, 20359 Hamburg, Germany

**Keywords:** tick-borne pathogens, birds, Republic of Moldova, surveillance, epidemiology, molecular diagnostics

## Abstract

Epidemiological knowledge on pathogens in ticks feeding on birds in Moldova is scarce. To reduce this gap of information, a total of 640 migrating and native birds of 40 species were caught from 2012 to 2015 and examined for the presence of ticks in the Republic of Moldova. Altogether, 262 ticks belonging to five tick species (*Ixodes ricunus* n = 245, *Ixodes frontalis* n = 12, *Haemaphysalis punctata* n = 2, *Hyalomma marginatum* n = 2 (only males), *Dermacentor marginatus* n = 1) were collected from 93 birds. Of these ticks, 250 (96%) were at the stage of a nymph and 9 at the stage of a larva (3%). One imago of *I. frontalis* and two imagoes of *Hy. marginatum* were found. Generally, ticks infested 14.1% of the assessed birds belonging to 12 species. DNA was extracted from individual ticks with subsequent PCR targeting *Rickettsia* spp., *Borrelia* spp. in general, as well as relapsing fever-associated *Borrelia* spp., in particular, *Anaplasma phagocytophilum*, *Neoehrlichia mikurensis, Babesia* spp. and *Coxiella burnetii*. The bird species *Turdus merula* showed the heaviest infestation with ticks and the highest incidence of infected ticks. Altogether, 32.8% of the assessed ticks (n = 86) were positive for one of the pathogens. DNA of *Borrelia* spp. was found in 15.2% (40/262) of the investigated ticks; in 7.6% of ticks (20/262), DNA of rickettsiae was detected; 6.9% (18/262) of the ticks were positive for *A. phagocytophilum* DNA; in 1.5% of the ticks (4/262), DNA of *Neoehrlichia mikurensis* was detected, followed by 1.5% (4/262) *Babesia microti* and 1.5% (4/262) *Borrelia miyamotoi.* Within the *B. burgdorferi* complex, *B. garinii* (n = 36) was largely predominant, followed by *B. valaisiana* (n = 2) and *B. lusitaniae* (n = 2). Among the detected *Rickettsia* spp., *R. monacensis* (n = 16), *R. helvetica* (n = 2) and *R. slovaca* (n = 1) were identified. In conclusion, the study provided some new information on the prevalence of ticks on birds in Moldova, as well as the presence of DNA of pathogens in the ticks. By doing so, it provided an additional piece in the puzzle of the global epidemiology of tick-transmitted infectious diseases from a geographic side from where respective surveillance data are scarce.

## 1. Introduction

Ticks are important vectors of animal and human pathogens. *Ixodes ricinus* can transmit important viral and bacterial pathogens, such as tick-borne encephalitis virus and *Borrelia burgdorferi* sensu lato, as well as other pathogens, such as *Babesia* spp., *Anaplasma phagocytophilum*, spotted fever group (SFG) rickettsiae, relapsing fever group *borreliae* and *Neoehrlichia mikurensis* [1,2].

To shortly summarize these pathogens, *Anaplasma phagocytophilum* causes anaplasmosis in dogs, horses, sheep and cattle, also known as tick-borne fever, and is the causative agent of human granulocytic anaplasmosis (HGA). The pathogen has been detected in *I. ricinus* in several European countries with prevalence ranging from 2% to 45% [3]. The first confirmed HGA case in Europe was diagnosed in Slovenia in 1997. Afterward, >100 HGA cases were reported mainly in Slovenia, Scandinavia and France [4,5,6,7]. Rickettsiae are bacterial pathogens transmitted by blood-sucking ectoparasites, such as ticks, fleas and lice. Members of the genus *Rickettsia*, pathogenic to humans, were traditionally classified into the spotted fever group (SFG), including symbionts transmitted by hard ticks, and the typhus group, including *Rickettsia typhi* [8]. Many different SFG *Rickettsiae* causing rickettsiosis in humans have been detected in Europe. The symptoms of rickettsiosis include fever, headache, rash, muscular pain and local lymphadenopathy [9]. *Babesia* spp. are protozoan pathogens that reside inside the erythrocytes of infected animals. Babesiae can be transmitted by ticks and, rarely, humans may be affected [10]. *B. microti* is primarily responsible for human babesiosis in North America and *B. divergens* in Europe. During the last 50 years, several hundred human clinical cases in North America and about 50 clinical cases in Europe have been reported, mostly in immunosuppressed patients [11]. In previous investigations on *I. ricinus* in several European countries, the species *B. divergens*, *B. microti* and *B.*
*venatorum* have been detected [11]. In addition, *B. duncani* (*Babesia* sp. WA1), *B. crassa-like*, *Babesia* sp. KO, *Babesia* sp. CN1 (*Babesia* sp. XXB/HangZhou) and *B. odocoilei* have been recently acknowledged as zoonotic species [12]. Lyme disease is a tick-transmitted multisystemic infection, which is caused by spirochetes of *Borrelia burgdorferi* sensu lato [2]. In recent years, increasing numbers of reports have been published on the detection of relapsing fever-associated borreliae, such as *Borrelia miyamotoi*, in hard ticks in central Europe and in associated human cases [13]. Further, human cases of *B. miyamotoi* have been reported in Russia, the United States, the Netherlands and Japan [14,15,16,17,18]. *Neoehrlichia mikurensis* is an emerging tick-borne pathogen causing a systemic inflammatory syndrome. In Europe, clinical symptoms caused by *N. mikurensis* infections have mainly been described in immunocompromised patients. The most frequent symptoms were fever, localized muscular pain and/or painful joints, as well as vascular and thromboembolic events [19,20,21].

By themselves, ticks are not highly mobile, and the most efficient way to expand their habitat is by hosts. Birds, due to their ability to fly, have the greatest influence on the resettlement of ticks [22,23]. In Europe, migratory birds generally host a number of tick species belonging to the genera *Ixodes*, *Haemaphysalis* and *Hyalomma* [24]. In Moldova, birds were found to be infested with several tick species in previous investigations, e.g., with *Ixodes ricinus*, *I. frontalis*, *Haemaphysalis punctata*, *I. lividus* and *Hyalomma marginatum* [25,26]. In eastern Europe, there are two major routes for bird migration, which merge into the eastern Mediterranean flyway, just on the border of the Republic of Moldova near the Danube Delta, which hosts more than 300 species of birds. Around half a million birds migrate through the territory of the Republic of Moldova each year [25,27]. Many of these travel from Africa and can therefore carry even tropical ticks, which, again, may host regionally abundant pathogens. Due to the continental climate in Moldova with hot summers and due to the general climate change, tropical ticks can at least survive during the summer season in Moldova [27].

There is hardly any information on the questions of which birds carry which ticks in Moldova, and which ticks host which pathogens to what extent. An assessment with historic ticks from the 1960s suggested the abundance of the abovementioned pathogens in low percentages [28]. Further, there is some, but still limited, information available from the neighboring countries [27,29,30]. However, precise information on the local presence of ticks and tick-borne pathogens is important in order to estimate the likeliness of respective diseases in humans and animals and in order to facilitate control measures if necessary. To contribute to the scarcely available epidemiological information, birds were caught in Moldova. Subsequently, ticks from those birds were collected, identified and screened by PCR for bacterial and protozoan pathogens comprising *Borrelia burgdorferi* sensu lato, relapsing fever group *Borrelia*, *Anaplasma phagocytophilum,* spotted fever group (SFG) rickettsiae, *Neoehrlichia mikurensis*, *Babesia* spp. and *Coxiella burnetii*. 

## 2. Materials and Methods

### 2.1. Sampling Sites

The collection of field material was carried out during the three field seasons of 2012–2015. The field material was collected in the territories of the Yagorlyk Reserve, Prutul de Jos Reserve, Codrii Reserve, Padurea Domneasca Reserve, around the city of Chisinau, in Chisinau Botanical Garden, as well as in the Badraji Vechi and Baltsata villages (Table 1, Figure 1). 

### 2.2. Sampling Strategy

During the field studies, birds were caught with the help of specialized nylon nets. A total of nine black nylon nets were applied. The nets were made in-house by trimming and redrawing of four Ecotone Inc. (Gdynia, Poland) nets from Poland (Ecotone Mist Net 716/12), which were converted from five pockets to three pockets, from 12 m to 6 m length and from a height of 2.5 m to 1.5 m.

Nine nets were installed at three different locations at each collection point. Three nets were located in dense vegetation or in the undergrowth; three were located on the border of forest plantations, and three were located in the bushes near human dwellings. For the permanent collection points, which were the Yagorlyk Nature Reserve and the Botanical Garden of Chisinau, every month, there were 3 full days of collecting; the interval between capturing days was from 6 to 10 days, depending on weather conditions. Collecting was not performed during rainy or windy days.

During each day, the nets were installed from sunrise to dawn and were checked approximately every hour in the afternoon, and every 30 min in the morning and in the evening. All birds were identified to the species level according to Cramp and Brooks [31]. If possible, gender and age were determined.

The head and neck of each bird were examined for the presence of ectoparasites. The feathers on the neck and head were checked with the help of entomological tweezers; special attention was paid to the favorite places of parasite concentration: the head, auricles, eyelids.

The captured birds were released immediately after the inspection. Ticks from birds were collected using specialized tweezers and placed in 70% ethanol using a separate 1.5 mL plastic tube for each bird. Ticks were identified in the laboratory under a stereomicroscope using the identification keys of Nosek (1972), Fillipova (1979, 1998) and Apanaskevich (2006, 2008) [32,33,34].

Investigations of the collected material by molecular genetic techniques were performed partially in Germany (Bernhard Nocht Institute, Hamburg; Ludwig-Maximilians-Universität, Munich; Friedrich-Loeffler-Institute, Department of Bacterial Infections and Zoonoses, Jena) and at the Center for Molecular Phylogeny in the Institute of Zoology, Chisinau, Moldova.

### 2.3. Nucleic Acid Extraction

In the laboratory, individual ticks were washed in distilled water and subsequently cut in two equal pieces. Then, one piece was stored in a freezer in a single tube at −20 °C for further putative analysis. The examined piece was cut into several pieces with a disposable scalpel, which were placed in 1.5 mL tubes in 100 μL phosphate-buffered saline (PBS). Samples were homogenized in a SpeedMill homogenizer (Jena, Germany) with the help of innuSPEED Ceramic beads Type P (2.4–2.8 mm) (Hannover, Germany). DNA was extracted individually from every tick using the QIAGEN DNAEasyBlood and Tissue Kit (QIAGEN, Hilden, Germany), according to the manufacturer’s instructions, with a minor modification as follows: the samples were incubated in ATL-buffer (30 mM Tris·Cl; 8 mM EDTA; 0.5% SDS) containing 1.25 µg/mL proteinase K for 60 min at 50 °C. The quantity and quality of the extracted DNA were evaluated with NanoDrop^®^ 2000 spectrophotometer analysis (NanoDropTechnologies, Wilmington, DE, USA).

### 2.4. Applied Molecular Pathogen Detection and Differentiation Approaches

For the amplification of the *B. burgdorferi* sensu lato DNA, a 5S-23S rDNA (intergenic spacer region (IGS)) fragment was used as a PCR target. The primers rrfA and rrlB with an expected PCR product size of 198 base pairs (bp) and the protocol were described by Richter and colleagues [35]. Band visualizing was conducted by electrophoresis on 2% agarose gels in Tris-acetate-EDTA-buffer (TAE), as well as staining with ethidium bromide (Sigma-Aldrich, Hamburg, Germany). Relapsing fever group borreliae were targeted with a hybridization probe-based real-time PCR, aiming at the 23S rDNA, as described [36], which was run on an AB7500fast cycler (Waltham, MA, USA). Positive samples were confirmed by gel PCR using another set of primers, as described by Assous and colleagues [37], targeting the *Borrelia*-specific *flaB* flagellin gene. Again, band visualization was based on electrophoresis on 1.2% agarose gel in TAE and staining with ethidium bromide.

Species of the *Rickettsia* spotted fever group were detected using a previously published hybridization probe-based real-time TaqMan PCR assay specific for a 74-bp fragment of the *gltA* gene [38] on RotorGene Q cyclers (Qiagen, Hilden, Germany). For positive samples, 2 additional conventional PCRs were added. First, a protocol based on the conventional PCR primers 120-M59 and 120-807, which amplify a 764-bp fragment of the rickettsial 135-kDa outer membrane protein B gene (*ompB*), was applied [39]. If an amplicon of the expected size was observed, the positive samples were also tested using another published protocol based on the CS1d and CS2d primers, which amplify 1254 bp of the *gltA* gene [40]. Amplicons obtained by conventional PCR were visualized by electrophoresis on 1% agarose gels in TAE and stained with ethidium bromide.

DNA eluates were screened for *A. phagocytophilum* using a real-time PCR targeting a 77 bp fragment of the *msp2* gene, as previously described [41]. To confirm positive *A. phagocytophilum* results, a nested PCR targeting a 497 bp region of the 16S rRNA gene was performed, as reported in the literature [42]. The detection of DNA of *Babesia* spp. was carried out with a conventional PCR targeting the *18S rRNA* gene [43]. For both the *A. phagocytophilum*-specific nested PCR and the *Babesia* spp.-specific conventional PCR, the amplicons were visualized in 2% agarose gel electrophoresis. Staining was performed with Gel Red (Biotium, Fremont, CA, USA) in Germany and with ethidium bromide in Moldova. For the detection of *N. mikurensis* DNA, a previously described real-time PCR targeting the *groEL* gene was applied [44]. A subset of 129 ticks, which were collected until summer 2014, was tested with a published real-time PCR protocol targeting the *icd* gene of *Coxiella burnetii* [45].

Applied oligonucleotides and further reaction details are indicated in the Appendix A Table A1.

PCR amplicons of the conventional PCRs targeting *Anaplasma*, *Babesia*, *Borrelia* and *Rickettsia* were purified with the GeneJetTM PCR Purification Kit (Thermo Fisher Scientific Inc., Waltham, MA, USA) and sent to Eurofins Genomics for bidirectional sequencing. Sequences were compared to sequences deposited in the NCBI GenBank, applying the BLASTn tool. Sequences of PCR products and those obtained from GenBank were edited using the BioEdit Sequence Alignment Editor (version 7.0.5.3; Hall, 1999; Raleigh, NC, USA). As sequencing was performed for diagnostic purposes only, no sequence information was deposited.

### 2.5. Statistical Analysis

Ginsberg’s coinfection index (Ic) was calculated to test for differences between the observed and expected co-infection rates. The Ic is positive when the number of co-infections is greater than expected. The significance of the index was assessed, applying the v2-test [46]. Kendall’s correlation coefficient (R) was used to evaluate a correlation between the total number of tested ticks and the total number of co-infected ticks.

### 2.6. Ethical Clearance

No bird was seriously injured during the study. The research was conducted pursuant to the Moldavian Code of Ethics. Since this work was carried out as part of a PhD thesis, the methodology and principles of the work were considered by a specialized commission of the State University Dimitrie Cantemir and the Institute of Zoology (Chisinau, Moldova). After a review of the provided research protocol, compliance with all regulatory requirements and local laws was confirmed. No regulatory or ethical issues were identified, and the protocol was approved.

## 3. Results

### 3.1. Detection of Ticks on Caught Birds

A total of 640 birds belonging to 40 species from 16 families were captured and examined for tick infestation during the study period. Details are provided in Table 2.

Ticks were found on 93 birds (35%). A total of 262 ticks were collected, belonging to five species (*I. ricinus* n = 245, *I. frontalis* n = 12, *Hae. punctata* n = 2, *Hy. marginatum* n = 2, *D. marginatus* n = 1), of which 250 (96%) were at the stage of a nymph and 9 at the stage of a larva (3%). Only one imago of *I. frontalis* and two imagoes of *Hy. marginatum* were found. The overall mean intensity and mean abundance of infestations were 2.81 and 0.62, respectively. The highest overall intensity of tick infestation was recorded at the sampling site of the Yagorlyk Reserve (Table 3 and Table 4).

### 3.2. Detection of Pathogen DNA in Collected Ticks and Coinfections

A total of 33% (86/262) (Table 5) of ticks collected from birds were associated with a positive result for one or more pathogens: 15.2% (40/262) of the ticks contained DNA of *Borrelia* spp., 7.6% (20/262) of rickettsiae, 6.9% (18/262) of the ticks were positive for DNA of *A. phagocytophilum*, 1.5% (4/262) for *N. mikurensis*, 1.5% (4/262) for *B. microti* and 1.5% (4/262) for *B. miyamotoi*. Half of the ticks (n = 129) were tested for the presence of *C. burnetii*, but no positive samples were recorded.

The 86 positive cases included 7 cases with co-detection (Table 6) of DNA of two pathogens. All pathogens were found in the preimaginal stages of *I. ricinus* with the exception of *R. slovaca*, which was found in a nymph of *Ixodes frontalis*.

In detail, *Borrelia* spp. DNA was detected in 15.2% (40/262) of the ticks. Four genospecies were identified by BLAST analysis of the amplified 198 bp fragment of the 5S-23S rDNA intergenic spacer region (IGS). *B. garinii* was most frequently identified with 100% sequence identity with previously deposited sequences (AY772205, GQ387030.1, JX909912.1, KU291355.1, KJ577538.1). *B. garinii* was found in *I. ricinus* preimaginal stages, collected from 25 birds comprising 16 blackbirds, 6 song thrushes and 3 common starlings. Both *B. lusitaniae* (n = 2) and *B. valaisiana* (n = 2) showed 99% identity with the deposited sequences HG798781.1 and CP009117.1, respectively. Those borreliae were found in *I. ricinus* nymphs collected from four blackbirds. *B. miyamotoi* (n = 4) sequences also reached 99% identity with the previously deposited sequences FJ874925.1 and CP010308.1. In 7.2% (19/262) of the ticks, rickettsial DNA was detected. The affected ticks were collected from common blackbirds and song thrushes. The DNA of rickettsial strains found in eight blackbirds (comprising fourteen *I. ricinus* nymphs) and two song thrushes (two *I. ricinus* nymphs and two *I. ricinus* larvae) showed 100% identity with *R. monacensis* (LN794217.1, AF141906.1, AF140706.1). The sequence from one tick sample collected from a blackbird (*I. ricinus* nymph) was 99% identical with *R. helvetica* (KP866150.1, KU310588.1), and another sequence from a *Rickettsia*-positive *I. frontalis* nymph collected from a song thrush was 100% homologous with a fragment of the full-genome sequence of the *R. slovaca* strain D-CWPP (GenBank accession no. CP003375). *Babesia* spp. DNA was demonstrated in four *I. ricinus* nymphs obtained from three blackbirds. Amplicons of the 452 bp fragment of the 18S rRNA gene were sequenced and showed 100% identity with deposited *B. microti* sequences (JQ886034.1 JQ886035.1 JQ886058.1). *A. phagocytophilum* DNA was detected in 28 *I. ricinus* nymphs collected from 14 birds comprising 8 blackbirds, 2 song thrushes, 3 common starlings and 1 European robin. Four sequences obtained from nested PCR were identical and showed 99% identity with previously deposited sequences (JX173651.1, JN181075.1, JN181063.1, HQ629911.1, AF136712.1). *N. mikurensis* DNA was amplified by real-time PCR from four *I. ricinus* nymphs collected from four blackbirds.

The overall level of co-detections based on the assessment of the 262 ticks was 2.7%. Ginsberg’s coefficient for co-infections (Ic) was +4.58 (*p* = 0.05). For all included settings, the Ic was positive (Table 6). Only in two territories (Prutul de jos, Badragii vechi) were there no cases of co-detections. Kendall’s correlation coefficient (R = 0.4, n = 262, *p* < 0.1) indicated that the total number of ticks was positively correlated with the number of *B. garinii* (as pathogen I) and *Rickettsia* spp. or *A. phagocytophylum* (as pathogen II) co-infected ticks. Most of the co-detections were identified at the study site of the Yagorlyk Reserve.

## 4. Discussion

The study was performed to assess the quantitative dimension of pathogens found in ticks collected from migratory birds in Republic of Moldova. Birds can serve as carriers of pathogens between countries or even continents by hosting ticks infected by diverse pathogenic agents. As migratory passerines usually migrate at high speed, they may easily transfer ticks over mountains and rivers, even seas. Reviews have summarized numerous studies proving the presence of multiple pathogens in ticks collected from migratory birds [47,48,49,50]. In Norway, for example, more than 9000 birds were tested for ticks with an infestation rate of 7.5% [22]. An extrapolation of these results suggests that millions of ticks are brought to Scandinavia annually by migrating birds with uncertain consequences for humans and animals. In most recent studies, however, only a small number of birds caught were infested with ticks, comprising 9% in Sweden [51], 8% in the UK [52] and 4% in Germany [53], with an average of two ticks per bird. The result of the present study with an infestation rate of 14.5% can be explained by the fact that when catching birds, nets were set on the preferred tracks of blackbirds.

As shown above, *Turdus merula* and *T. philomelos* were the species with the highest infestation rate. These species are ground feeders, which greatly increases their contact with ticks. In contrast, most species of birds collected during the study were not infested by ticks. The majority of these numerous birds (*Coccothraustes coccothraustes*, *Fringilla coelebs*, *Carduelis chloris*) comprised non-ground feeder species. So, it is likely that feeding preference is a major factor affecting the frequency of tick infestation on birds. This hypothesis is partially confirmed by a recently published study; however, the phenomenon seems to largely depend on the tick species [53]. The variability of the environmental conditions of Moldova is associated with the presence of passerine birds in different seasonal periods. The wide species spectrum and the high number of birds cause a high probability of the introduction of pathogens, which are brought by the ticks feeding on the birds, and thus, of the formation of secondary natural foci.

In this study, five tick species parasitizing on birds were collected. Four of these are endemic in the Republic of Moldova, but *Hy. marginatum* has not been reported in Moldova since the 1970s. Fifty years ago, *Hy. marginatum* were numerous in Moldova but could not reproduce in the environment due to harsh winters. They parasitized on cattle and survived the winters in barns [54]. Presumably, they arrived in Moldova in the 1960s from Bulgaria along with cattle. In the present study, *Hy. marginatum* were collected from *Acrocephalus arundinaceus* birds in April at the southern point of the country (reserve Prutul de Jos). *A. arundinaceus* is considered a long-distance migrant. Both specimens of *Hy. marginatum* collected from birds were male. Females of this species are less likely to be attached to a bird for 5–7 migratory days because females are engorged faster and can reach a bigger size. This can cause physical discomfort for birds, especially in a case of multiple ticks parasitizing on a single bird. Additionally, in a case of multiple parasitism and depending on the type of bird, the loss of blood can lead to physical weakening, which slows migration down, as suggested by Møller and Erritzøe’s [55]. *Hy. marginatum* serve as vectors for pathogens and can transmit Crimean hemorrhagic fever virus, which was not in the focus of this study. In case of an increase in the average annual temperature, an increase in aridity and mild winters, *Hy. marginatum* might become resident in the territory of the Republic of Moldova again. Therefore, the detection of these ticks on the territory of the Republic of Moldova indicates a need for further research and monitoring activities.

In total, from the beginning of the 1950s of the 20th century to the present time, 23 species of ixodids have been registered in Moldova. Of these, seven species were collected from birds, which included *I. ricinus*, *I. lividus*, *I. frontalis*, *I. crenulatus*, *Haemaphysalis punctata*, *Hae. caucasica* and *Hae. concinna*. [25,26,56]. Thereby, *Hae. caucasica* and *Hae. concinna* were collected in only a few incidents; *I. lividus* is a highly specialized parasite of *Riparia riparia* in the Republic of Moldova; *I. crenulatus* was found parasitizing only on *Oenanthe oenanthe* and on pigeons. *I. frontalis* was mainly found on birds of the Turdidae family, but in very small amounts, i.e., 10 infected birds with *I. frontalis* among 900 surveyed ones in 1985 (1.1%). This almost coincides with the results of the present study, in which 6 birds out of 640 (0.94%) were infested with *I. frontalis*. In the present study, a case of finding *D. marginatus* nymph on *Dendrocopos syriacus* was registered. Since *Dendrocopos syriacus* is not a ground feeder, and *D. marginatus* is not ornithophilic ectoparasite, we consider this case to be a rare casuistry and a coincidence.

Focusing on pathogen detection rates within the assessed ticks, in one-third of the ticks, the DNA of one or more pathogens was found. This level of detection is twice as high as the rate recorded in ticks collected from vegetation in the same area [57]. In particular, the detection rate of ticks with DNA of the pathogen *A. phagocytophilum* was found to be high in Moldova, at 7.2%, which is slightly higher than the average in ticks collected from vegetation in neighboring Ukraine, for which a detection rate of 5.2% has been reported [29]. The prevalence of rickettsiae and borreliae was found to be similar as previously recorded for ticks collected from vegetation [57] and was 7.6% and 15.3%, respectively. The prevalence of *R. monacensis* was significantly higher than of *R. helvetica*, which is in contrast to a previous assessment in northwestern Russia [24]. In this study, the presence of *B. miyamotoi* was first described in ticks from Republic of Moldova.

The seasonal distribution of different groups of pathogenic microorganisms varied greatly. Most ticks with the DNA of *A. phagocytophilum* were collected in the spring (90% of cases), while *Borrelia* spp.-infected nymphs were more often found in the fall (75% of cases). *Rickettsia*-positive ticks were detected at all times of the year, with the exception of winter.

## 5. Conclusions

The study has shown that birds migrating through Moldova carry ticks infected with a high diversity of pathogens. Further, this is the first record on the occurrence of *B. miyamotoi* in the Republic of Moldova and the first regional record of collecting *Hy. marginatum* from migratory birds. For the Republic of Moldova, this is the first large-scale long-term study of ticks collected from birds; this study is filling the gap in knowledge about the circulation of tick-borne pathogens in the wild for a given region. The data obtained are of interest to the Public Health Centers as contributing to the predictions of the epidemiological situation.

## Figures and Tables

**Figure 1 microorganisms-10-01111-f001:**
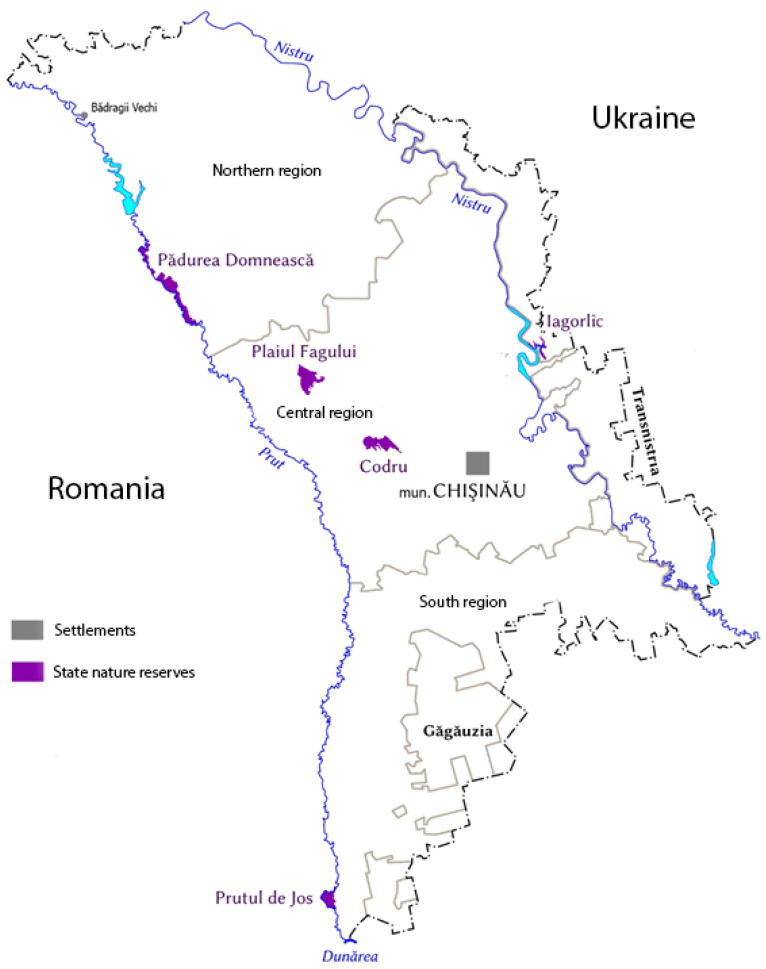
Visualization of the collection sites.

**Table 1 microorganisms-10-01111-t001:** Collection points during the survey period 2012–2015.

Collecting Point	№	Place of the Bird Collection	Geographical Coordinates	Collection Period (Month, Year)
Urban areas	1	Chisinau Botanical Garden	46°58′23.0″ N 28°53′08.7″ E	III–XI 2012–2015
	2	Chisinau, Park Riscani	47°02′53.4″ N 28°52′32.4″ E	III–XI 2012–2015
	3	Durlesti, outskirts of the city	47°01′40.4″ N 28°44′36.9″ E	IV–VI 2012–2014
Agrocenoses	4	Badragii Vechi village	48°01′54.4″ N 27°06′40.3″ E	VI 2012, VI 2014
	5	Badragii Vechi village	48°01′51.4″ N 27°06′38.8″ E	VI 2012, VI 2014
	6	Baltsata village	47°02′53.9″ N 29°02′28.1″ E	V–VI 2014
Reserve zones	7	Reserve Yagorlyk	47°23′03.2″ N 29°10′12.0″ E	III–VI 2012–2015, IX–XI 2012–2015
	8	Reserve Yagorlyk	47°23′01.2″ N 29°10′42.3″ E	III–VI 2012–2015, IX–XI 2012–2015
	9	Reserve Yagorlyk	47°23′07.1″ N 29°10′28.3" E	III–VI 2012–2015, IX–XI 2012–2015
	10	Reserve Prutul de Jos	45°35′30.6″ N 28°09′37.9″ E	V 2014, V 2015
	11	Reserve Prutul de Jos	45°35′25.0″ N 28°09′35.0″ E	IX 2015
	12	Reserve Padurea Domneasca	47°36′22.1″ N 27°23′41.0″ E	IV 2012, V 2015
	13	Reserve Padurea Domneasca	47°36′22.1″ N 27°23′41.0″ E	X 2012, IX 2014
	14	Reserve Plaiul Fagului	47°17′40.6″ N 28°01′50.3″ E	VI 2014
	15	Reserve Plaiul Fagului	47°18′60.0″ N 28°02′30.8″ E	VI 2015
	16	Reserve Codrii	47°03′27.1″ N 28°33′36.7″ E	IV–V 2012, IV–V 2014
	17	Reserve Codrii	47°03′25.7″ N 28°33′37.3″ E	IX–XI 2014, V 2015

**Table 2 microorganisms-10-01111-t002:** Birds collected during the study period.

Collection Places	Iagorîc	Plaiul Fagului	Codrii	Prutul de Jos	Pădurea Domneasca	Mun. Chișinău	Bădragii Vechi	Total
*Passer domesticus*	15	7	4	5	9	20	18	78
*Turdus merula*	42	4	6	3	2	15	6	78
*Sturnus vulgaris*	9	5	6	7	4	15	20	66
*Erithacus rubecula*	22	6	9	4	3	5	3	52
*Turdus philomelos*	21	6	3	5	7	9	1	52
*Parus major*	13	1	1	2	3	21	3	44
*Coccothraustes coccothraustes*	13	6	1	2	1	12	3	38
*Carduelis chloris*	11	3	3	2	2	1	1	23
*Fringilla coelebs*	6	5	4	6	0	4	1	26
*Luscinia luscinia*	0	1	0	2	0	13	0	16
*Lanius collurio*	2	1	0	0	1	0	10	14
*Passer montanus*	5	2	0	0	2	3	2	14
*Dendrocopos syriacus*	6	0	1	0	0	4	1	12
*Garrulus glandarius*	4	0	2	0	1	2	1	10
*Sylvia atricapilla*	3	1	0	0	4	2	0	10
*Acrocephalus arundinaceus*	0	0	0	8	0	0	0	8
*Cyanistes caeruleus*	6	0	0	2	0	0	0	8
*Lanius minor*	2	0	1	0	1	1	3	8
*Oriolus oriolus*	0	1	0	0	1	2	4	8
*Pica pica*	2	0	0	0	0	6	0	8
*Dendrocopos major*	3	0	2	0	0	1	1	7
*Emberiza citronella*	2	0	0	2	1	1	0	6
*Picus canus*	2	0	0	0	1	3	0	6
*Prunella modularis*	0	1	2	2	1	0	0	6
*Sitta europaea*	2	3	1	0	0	0	0	6
*Anthus trivialis*	2	0	0	0	0	2	0	4
*Hirundo rustica*	0	0	0	4	0	0	0	4
*Phoenicurs ochruros*	2	0	0	0	0	2	0	4
*Accipiter nisus*	3	0	0	0	0	0	0	3
*Emberiza schoeniclus*	1	0	0	2	0	0	0	3
*Turdus pilaris*	3	0	0	0	0	0	0	3
*Motacilla alba*	1	0	0	0	0	0	1	2
*Accipiter gentilis*	2	0	1	0	0	0	1	4
*Alcedo atthis*	0	0	0	2	0	0	0	2
*Corvus frugilegus*	0	0	0	0	0	2	0	2
*Carduelis spinus*	1	0	0	0	0	0	0	1
*Dendrocopos minor*	1	0	0	0	0	0	0	1
*Jynx torquilla*	1	0	0	0	0	0	0	1
*Phoenicurus phoenicurus*	0	0	0	0	0	1	0	1
*Phylloscopus trochilus*	1	0	0	0	0	0	0	1
**Total**	**209**	**53**	**47**	**60**	**44**	**147**	**80**	**640**

**Table 3 microorganisms-10-01111-t003:** Distribution of birds and ticks by collection site.

	*Yagorlyk*	*Chisinau and suburbs*	*Plauil Fagului*	*Codrii*	*Prutul de Jos*	*Padurea Domneasca*	*Vilages* *(B. vechi and Baltsata)*	Total
*Birds examined*	209	147	53	47	60	44	80	**640**
*Infested birds*	40	25	14	7	4	0	3	**93**
*Ticks collected*	165	42	19	21	10	0	5	**262**
*Ticks in which DNA of at least one of the pathogenic agents was found*	39	28	9	7	2	0	1	**86**

**Table 4 microorganisms-10-01111-t004:** Species composition and tick infestation of birds caught during the study in Moldova. Mean intensity = number of ticks/number of infested birds. Mean abundance = number of ticks/number of birds. L = larva. N = nymph. I = imago.

Bird Species (Collected/Infested)	Number of Birds Infested with Ticks (In Brackets—Number of Collected Ticks)								Prevalence (%)	Ticks Collected	Mean Intensity	Mean Abundance
	*I. ricinus*		*I. frontalis*			*Hae. punctata*	*D. marginatus*	*Hy. marginatum*				
	L	N	N	I♀	L	N	N	I ♂				
*Turdus merula* 78/46		44 (164)	4 (6)		2 (2)	2 (2)			59	174	3.78	2.23
*Turdus philomelos* 52/10	4 (4)	8 (22)	2 (3)	1 (1)					18	32	3.2	0.59
*Sturnus vulgaris* 66/10		10 (16)							15	16	1.6	0.26
*Luscinia luscinia* 16/8		8 (8)							50	8	1	0.5
*Erithacus rubecula* 52/4		4 (4)							8	4	1	0.07
*Parus major* 44/2	2 (3)	2 (9)							5	12	6	0.27
*Anthus trivialis* 4/2		2 (4)							50	4	2	1
*Corvus frugilegus* 2/2		2 (2)							100	2	1	1
*Passer domesticus* 78/4		4 (4)							5	4	1	0.05
*Sylvia atricapilla* 10/2		2 (2)							20	2	1	0.2
*Dendrocopos syriacus* 12/1		1 (1)					1 (1)		8.4	1	1	0.08
*Acrocephalus arundinaceus* 8/2		2 (2)						2 (2)	25	2	1	0.25
**Total** **422/93**									**22.1**	**262**	**2.81**	**0.62**

**Table 5 microorganisms-10-01111-t005:** Pathogens’ DNA in ticks collected from birds in Republic of Moldova. N = nymph. L = larva.

Species of Ticks	Pos/No. Total	The Number of Cases of DNA Detection of the Pathogenic Agents									
		*B. microti*	*N. mikurensis*	*A. phagocytophilum*	*B. miyamotoi*	*R. monacensis*	*R. slovaca*	*R.* *helvetica*	*B. garinii*	*B.* *valaisiana*	*B. lusitaniae*
*I. ricinus* N	82/239	4	4	16	4	19		2	35	2	2
*I. ricinus* L	3/7			2		1			1		
*I. frontalis* N	1/9						1				

**Table 6 microorganisms-10-01111-t006:** Co-detection rates in ticks collected from birds in Moldova.

Tick Species	Site	Number of Ticks Examined	Number of Ticks with Co-Detection (%)	Co-Detection Index	Co-Detection Type	Bird Species
*I. ricinus*	Iagorlîc	165	3 (1.8)	+1.37 *	B.g./R.m	*T. merula*
					B.g./R.m	*T. philomelos*
					B.g./A.p	*T. merula*
	mun. Chișinău	42	2 (4.7)	+2.5 *	B.g./R.m	*T. merula*
					B.g./A.p	*T. merula*
	Plaiul fagului	19	1 (5.3)	+12.0	R.m./A.p	*T. merula*
	Codrii	21	1 (4.7)	+17.8	B.g./R.m	*T. merula*
	Prutul de jos	10	0			
	Badragii vechi	5	0			
Total		262	7 (2.67)	+4.58 *		

B.g.—*B. garinii*; A.p.—*A. phagocytophylum*; R.m.—*R. monacensis*; * Types of co-detection with statistical significance (*p* < 0.05).

## Data Availability

All relevant data are provided in the manuscript. Raw data can be made available on reasonable request.

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
