# Peer review of "Prevalence of Bacterial and Protozoan Pathogens in Ticks Collected from Birds in the Republic of Moldova"

_microorganisms, 2022, doi:10.3390/microorganisms10061111_

Round 1

Reviewer 1 Report

  1. The abstract must also reflect the significance of the study.
  2. Line 57 do elaborate why the disease is most likely to be under diagnosed, the bio burden in the bodily fluids is not much.
  3. It should be clear through the writing though there are various report of tick born pathogens and their transmissions throughout the world, why it is important to study the same in republic of Moldova?
  4. Lines 143 to 146 are confusing please rephrase
  5. If B. burgdoferi were cultured do mention that.
  6. Based on the materials present in discussion do try a PCA plot, it would substantiate your hypothesis 

Author Response

Point 1: The abstract must also reflect the significance of the study.

Response 1: As now clarified more explicitely, the study was meant to provide an additional piece in the puzzle of the global epidemiology of tick transmitted infectious disease from a geographic side from where respective surveillance data are scarce.

Point 2: Line 57 do elaborate why the disease is most likely to be under diagnosed, the bio burden in the bodily fluids is not much.

Response 2: We decided to delete this phrase, because the detailed explanation of the reasons for under diagnositng varies from country to country and it is not possible to briefly give a satisfactory explanation,

Point 3: It should be clear through the writing though there are various report of tick borne pathogens and their transmissions throughout the world, why it is important to study the same in republic of Moldova?

Response 3: The importance has been addressed in the last paragraph of the introduction. As stated there, respective surveillance data from Moldova are decades old, while precise information on the local presence of ticks and tick-borne pathogens is important in order to estimate the likeliness of respective diseases in humans and animals and in order to facilitate control measures if necessary.

Point 4: Lines 143 to 146 are confusing please rephrase

Response 4: The description was changed

from
“The feathers on the neck and head were “combed” with entomological tweezers against the growth of the feathers, special attention was paid to the favorite places of parasite concentration: the head, auricles, eyelids, the lower soft part of the beak and the sides of the beak, neck and under the wings.”
to
“The feathers on the neck and head were checked with the help of entomological tweezers. Special attention was paid to the favorite places of parasite concentration: the head, auricles, eyelids.”

Point 5: If B. burgdoferi were cultured do mention that.

Response 5: Indeed, B. burgdoferi were not grown in culture during this study.

Point 6: Based on the materials present in discussion do try a PCA plot, it would substantiate your hypothesis 

Response 6: We have abstained from this type of visualization for various reasons. First, the number of detected pathogens was quite small. Second, nearly all pathogens were identified in I. ricinus ticks with few examples. If the bird species and the various geographical side would have been included, the various clusters would have become very small. So, we feel that visualization by PCA plotting would not have increased the clarity and we respectfully ask the editor to accommodate our decision.

Reviewer 2 Report

This is well designed and written epidemiological study. I have only two comments: about number of collected ticks and one pathogen species name.

Both in the abstract and results section the authors have written about 262 collected ticks. However, 246 I. ricinus + 12 I. frontalis + 2 H. punctata + 2 H. marginatum + 1 D. marginatus equals 263 ticks. Moreover, 252 nymphs + 9 larvae + 3 adults (1 I. frontalis and 2 H. marginatum) equals 264 ticks. Thus, which number of ticks is proper? 262, 263 or 264?

Babesia EU1 is now Babesia venatorum.

Author Response

Point 1: Both in the abstract and results section the authors have written about 262 collected ticks. However, 246 I. ricinus + 12 I. frontalis + 2 H. punctata + 2 H. marginatum + 1 D. marginatus equals 263 ticks. Moreover, 252 nymphs + 9 larvae + 3 adults (1 I. frontalis and 2 H. marginatum) equals 264 ticks. Thus, which number of ticks is proper? 262, 263 or 264?

Response 1: Thank you for identifying this error, we have fixed it in text. The corrected numbers are Ixodes ricinus n= 245, nymphs n=250. In the table 4, all numbers had been correct, the mistake had been made in the text only.

Point 2: Babesia EU1 is now Babesia venatorum.

Response 2: The nomenclature has been adapted in the manuscript.

Reviewer 3 Report

The paper is a well-structured study about the occurrence of tick borne pathogens in Ixodida collected from birds in Moldova. The topic is of interest and data are relevant. Before the acceptance the Authors should make some modifications and add some information about zoonotic Babesia spp. Please, see below

line 46 and everywhere - omit ticks after the proper name

line 48 - please omit (B.)

line 51 - please, omit bacterium after the proper name

line 55 - please omit diagnostically

line 70 - Babesia sp. EU1 is now recognized as Babesia venatorum

line 71, 86 and elsewhere (mostly in results section) - after the first mention please abbreviate the genus' name

line 84 - and not italics

table A1 - please, modify ssp in spp and write not in italics

please, see the review papers

10.3390/pathogens11030298

10.3390/vetsci8120334

Author Response

Point 1: make some modifications and add some information about zoonotic Babesia spp.

Response 1: In the introduction, we have now stated that  B. duncani (Babesia sp. WA1), B. crassa-like, Babesia sp. KO, Babesia sp. CN1 (Babesia sp. XXB/HangZhou) and B. odocoilei have been recently acknowledged as zoonotic species. The respective refence (Ebani et al., 2021) has be added.

Point 2: line 46 and everywhere - omit ticks after the proper name.

Response 2: The inappropriate wording has been corrected throughout the document,

Point 3: line 48 - please omit (B.)

Response 3: Fixed

Point 4: please, omit bacterium after the proper name

Response 4: Fixed

Point 5: please omit diagnostically

Response 5: Fixed

Point 6: line 70 - Babesia sp. EU1 is now recognized as Babesia venatorum

Response 6: Fixed

Point 7: line 71, 86 and elsewhere (mostly in results section) - after the first mention please abbreviate the genus' name

Response 7: Fixed

Point 8: line 84 - and not italics

Response 8: Fixed

Point 9: table A1 - please, modify ssp in spp and write not in italics

Response 9: Fixed